# The Synthesis of XNOR Recurrent Neural Networks with Stochastic Logic

**Arash Ardakani, Zhengyun Ji, Amir Ardakani, Warren J. Gross**
Department of Electrical and Computer Engineering, McGill University, Montreal, Canada
{arash.ardakani, zhengyun.ji, amir.ardakani}@mail.mcgill.ca
warren.gross@mcgill.ca

## Abstract

The emergence of XNOR networks seek to reduce the model size and computational cost of neural networks for their deployment on specialized hardware requiring real-time processes with limited hardware resources. In XNOR networks, both weights and activations are binary, bringing great benefits to specialized hardware by replacing expensive multiplications with simple XNOR operations. Although XNOR convolutional and fully-connected neural networks have been successfully developed during the past few years, there is no XNOR network implementing commonly-used variants of recurrent neural networks such as long short-term memories (LSTMs). The main computational core of LSTMs involves vector-matrix multiplications followed by a set of non-linear functions and element-wise multiplications to obtain the gate activations and state vectors, respectively. Several previous attempts on quantization of LSTMs only focused on quantization of the vector-matrix multiplications in LSTMs while retaining the element-wise multiplications in full precision. In this paper, we propose a method that converts all the multiplications in LSTMs to XNOR operations using stochastic computing. To this end, we introduce a weighted finite-state machine and its synthesis method to approximate the non-linear functions used in LSTMs on stochastic bit streams. Experimental results show that the proposed XNOR LSTMs reduce the computational complexity of their quantized counterparts by a factor of $86\times$ without any sacrifice on latency while achieving a better accuracy across various temporal tasks.

## 1 Introduction

Recurrent neural networks (RNNs) have exhibited state-of-the-art performance across different temporal tasks that require processing variable-length sequences such as image captioning [1], speech recognition [2] and natural language processing [3]. Despite the remarkable success of RNNs on a wide range of complex sequential problems, they suffer from the exploding gradient problem that occurs when learning long-term dependencies [4, 5]. Therefore, various RNN architectures such as long short-term memories (LSTMs) [6] and gated recurrent units (GRUs) [7] have emerged to mitigate the exploding gradient problem. Due to the prevalent use of LSTMs in both academia and industry, we mainly focus on the LSTM architecture in this work. The recurrent transition in LSTM is performed in two stages: the first stage performing gate computations and the second one performing state computations. The gate computations are described as

$$\mathbf{f}_t = \sigma\left(\mathbf{W}_{fh}\mathbf{h}_{t-1} + \mathbf{W}_{fx}\mathbf{x}_t + \mathbf{b}_f\right), \ \mathbf{i}_t = \sigma(\mathbf{W}_{ih}\mathbf{h}_{t-1} + \mathbf{W}_{ix}\mathbf{x}_t + \mathbf{b}_i),$$
$$\mathbf{o}_t = \sigma(\mathbf{W}_{oh}\mathbf{h}_{t-1} + \mathbf{W}_{ox}\mathbf{x}_t + \mathbf{b}_o), \ \ \mathbf{g}_t = \tanh(\mathbf{W}_{gh}\mathbf{h}_{t-1} + \mathbf{W}_{gx}\mathbf{x}_t + \mathbf{b}_g), \tag{1}$$

where $\{\mathbf{W}_{fh}, \mathbf{W}_{ih}, \mathbf{W}_{oh}, \mathbf{W}_{gh}\} \in \mathbb{R}^{d_h \times d_h}$, $\{\mathbf{W}_{fx}, \mathbf{W}_{ix}, \mathbf{W}_{ox}, \mathbf{W}_{gx}\} \in \mathbb{R}^{d_x \times d_h}$ and $\{\mathbf{b}_f, \mathbf{b}_i, \mathbf{b}_o, \mathbf{b}_g\} \in \mathbb{R}^{d_h}$ denote the recurrent weights and bias. The input vector $\mathbf{x} \in \mathbb{R}^{d_x}$ denotes input temporal

features whereas the hidden state $\mathbf{h} \in \mathbb{R}^{d_h}$ retains the temporal state of the network. The logistic sigmoid and hyperbolic tangent functions are denoted as $\sigma$ and $\tanh$, respectively. The updates of the LSTM parameters are regulated through a set of gates: $\mathbf{f}_t$, $\mathbf{i}_t$, $\mathbf{o}_t$ and $\mathbf{g}_t$. The state computations are then performed as

$$\mathbf{c}_t = \mathbf{f}_t \otimes \mathbf{c}_{t-1} + \mathbf{i}_t \otimes \mathbf{g}_t, \ \mathbf{h}_t = \mathbf{o}_t \otimes \tanh(\mathbf{c}_t), \tag{2}$$

where the parameter $\mathbf{c} \in \mathbb{R}^{d_h}$ is the cell state. The operator $\otimes$ denotes the Hadamard product.

The first computational stage of LSTM is structurally similar to a fully-connected layer as it only involves several vector-matrix multiplications. Therefore, LSTMs are memory intensive similar to fully-connected layers [8]. LSTMs are also computationally intensive due to their recursive nature [9]. These limitations make LSTM models difficult to deploy on specialized hardware requiring real-time processes with inferior hardware resources and power budget. Several techniques have been introduced in literature to alleviate the computational complexity and memory footprint of neural networks such as low-rank approximation [10], weight/activation pruning [11, 12, 13, 14] and quantization [15, 16, 17]. Among these solutions, quantization methods specifically binarization methods bring significant benefits to dedicated hardware since they reduce the required memory footprint and implementation cost by constraining both weights and activations to only two values (i.e., -1 or 1) and replacing multiplications with simple XNOR operations, respectively [17]. As a result, several attempts were reported in literature to binarize LSTM models during the past few years [18, 19, 20]. However, all the existing methods only focused on the gate computations of LSTMs by binarizing either the weights or both the weights and the hidden vector $\mathbf{h}$ while retaining the state computations in full precision (FP). Although the recurrent computations of LSTM models are dominated by the gate computations, using full-precision multipliers are inevitable for the state computations when designing dedicated hardware for LSTM models, making the existing binarized LSTM models unsuitable for embedded systems with limited hardware resources and tight power budget. It is worth mentioning that a full-precision multiplier requires $200\times$ more Xilinx FPGA slices than an XNOR gate [17]. Therefore, an XNOR-LSTM model that can perform the multiplications of both the gate and the state computations using XNOR operations is missing in literature.

In this paper, we first extend an existing LSTM model with binary weights to binarize the hidden state vector $\mathbf{h}$. In this way, the multiplications of the gate computations can be performed using XNOR operations. We then propose a method to binarize the state computations using stochastic computing (SC) [21]. More precisely, we show that the binarized weights and the hidden state vector $\mathbf{h}$ can be represented as stochastic bit streams, allowing us to perform the gate computations using stochastic logic and to implement the non-linear activation functions (i.e., the sigmoid and hyperbolic tangent functions) using finite-state machines (FSMs). We also introduce a new FSM topology and its synthesis method to accurately approximate the nonlinear functions of LSTM. We show that the proposed FSM outputs a binary stream that its expected value is an approximation to the nonlinear activation functions of LSTMs. Ultimately, we use the binary streams generated by the FSMs to replace the full-precision multipliers required for the state computations with XNOR gates, forming an XNOR-LSTM model.

## 2 Related Work

In the binarization process, the full-precision weight matrix $\mathbf{W} \in \mathbb{R}^{d_I \times d_J}$ is estimated using a binary weight matrix $\mathbf{W}^b \in \{-1, 1\}^{d_I \times d_J}$ and a scaling factor $\alpha \in \mathbb{R}^+$ such that $\mathbf{W} \approx \alpha \mathbf{W}^b$. In [15], the sign function was used as the transformation function to obtain the binary weight matrix (i.e., $\mathbf{W}^b = \text{sign}(\mathbf{W})$) while using a fixed scaling factor for all the weights. Lin *et al.* [22] introduced a ternarization method to reduce the accuracy loss of the binarization process by clamping values hesitating to be either 1 or -1 to zero. Some methods [23, 24] were then proposed to improve upon the ternarization method by learning the scaling factor $\alpha$. Zhou *et al.* [25] proposed a method that quantizes weights, activations and gradients of neural networks using different bitwidths. Rastegari *et al.* [16] and Lin *et al.* [17] proposed binary neural networks (BNNs) in which both weights and activations of convolutional neural networks (CNNs) are represented in binary. Despite the great performance of the aforementioned works in quantization of CNNs, they fail to work well on RNNs [26]. As a result, recent studies mainly attempted to quantize RNNs in particular LSTMs.

Hou *et al.* [19] introduced the loss-aware binarization method (LAB) that uses the proximal Newton algorithm to minimize the loss w.r.t the binarizied weights. The LAB method was further extended

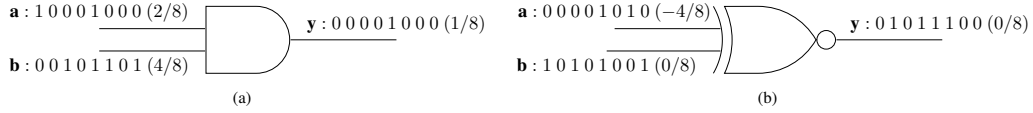

Figure 1: Stochastic multiplications using bit-wise operations in (a) unipolar and (b) bipolar formats.

in [27] to support different bitwidths for the weights. Both of these methods were tested on LSTM models performing character-level language modeling experiments. Xu *et al.* [18] presented the alternating multi-bit quantization (AMQ) method that uses a binary search tree to obtain optimal quantization coefficients for LSTM models. Wang *et al.* [26] proposed a ternary RNN, called HitNet, which exploits a hybrid of different quantization methods to quantize the weights and the hidden state vector based on their statistical characteristics. Both HitNet and alternating multi-bit quantization method were tested on RNNs performing word-language modeling experiments. Recently, Ardakani *et al.* [20] leveraged batch normalization in both the input-to-hidden and the hidden-to-hidden transformations of LSTMs to binarize/ternarize the recurrent weights. This method was tested on various sequential tasks, such as sequence classification, language modeling, and reading comprehension. While all the aforementioned approaches successfully managed to quantize the weights and the hidden state vector (i.e., the gate computations) of LSTM models, the state computations were retained in full precision. More precisely, no attempt was reported to binarize both the gate and state computations of LSTMs. Motivated by this observation, we propose the first XNOR-LSTM model in literature, performing all the recurrent multiplications with XNOR operations.

## 3 Preliminaries

### 3.1 Stochastic Computing

Stochastic computing is a well-known technique to obtain ultra low-cost hardware implementations for various applications [28]. In SC, continuous values are represented as sequences of random bits, allowing complex computations to be computed by simple bit-wise operations on the bit streams. More precisely, the statistics of the bits determine the information content of the stream. For example, a real number $a \in [0, 1]$ is represented as the sequence $\mathbf{a} \in \{0, 1\}^l$ in SC's *unipolar* format such that

$$\mathbb{E}[\mathbf{a}] = a, \tag{3}$$

where $\mathbb{E}[\mathbf{a}]$ and $l$ denote the expected value of the Bernoulli random vector $\mathbf{a}$ and the length of the sequence, respectively. Another well-known SC's representation format is the *bipolar* format where $a \in [-1, 1]$ is represented as

$$\mathbb{E}[\mathbf{a}] = (a + 1)/2. \tag{4}$$

To represent any real number using these two formats, we need to scale it down to fit within the appropriate interval (i.e., either $[0, 1]$ or $[-1, 1]$). It is worth mentioning that the stochastic stream $\mathbf{a}$ is generated using a linear feedback shift register (LFSR) and a comparator in custom hardware [28], referred to as stochastic number generator (SNG).

#### 3.1.1 Multiplication and Addition in SC

Multiplication of two stochastic streams of $\mathbf{a}$ and $\mathbf{b}$ in the unipolar format is performed as

$$\mathbf{y} = \mathbf{a} \cdot \mathbf{b}, \tag{5}$$

where "$\cdot$" denotes the bit-wise AND operation, $\mathbb{E}[\mathbf{y}] = \mathbb{E}[\mathbf{a}] \times \mathbb{E}[\mathbf{b}]$ if and only if the input stochastic streams (i.e, $\mathbf{a}$ and $\mathbf{b}$) are independent. However, this multiplication in the bipolar format is computed using an XNOR gate as

$$\mathbf{y} = \mathbf{a} \odot \mathbf{b}, \tag{6}$$

where "$\odot$" denotes the bit-wise XNOR operation. Similarly, if the input sequences are independent, we have

$$2 \times \mathbb{E}[\mathbf{y}] - 1 = (2 \times \mathbb{E}[\mathbf{a}] - 1) \times (2 \times \mathbb{E}[\mathbf{b}] - 1). \tag{7}$$

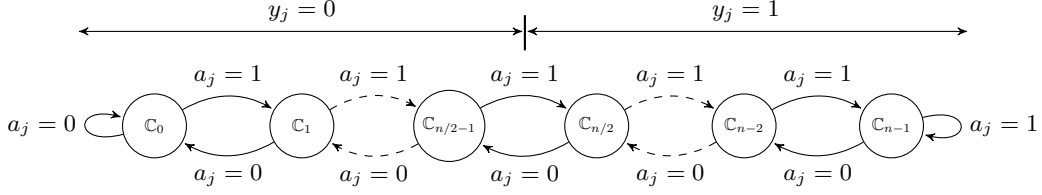

Figure 2: State transition diagram of the FSM implementing $\tanh$ where $a_j$ and $y_j$ denote the $j^{\text{th}}$ entry of the input stream $\mathbf{a} \in \{0,1\}^l$ and the output stream $\mathbf{y} \in {0,1}^l$ for $j \in \{1,2,\ldots,l\}$, respectively.

Figure 1 shows an example of a multiplication in stochastic domain using both the unipolar and bipolar formats. Since stochastic numbers are represented as probabilities falling into the interval of $[0,1]$ in the unipolar format, additions in SC are performed using the scaled adder that fits the result of the addition into the $[0,1]$ interval [28]. Additions of two stochastic streams of $\mathbf{a}$ and $\mathbf{b}$ is computed by

$$\mathbf{y} = \mathbf{a} \cdot \mathbf{c} + \mathbf{b} \cdot (1 - \mathbf{c}), \tag{8}$$

where the signal $\mathbf{c}$ is a stochastic stream with a probability of $0.5$ (i.e., $\mathbb{E}[\mathbf{c}] = 0.5$). The scaled adder is implemented using a multiplexer in which the stream $\mathbf{c}$ is used as its selector signal. The aforementioned discussion on the stochastic addition also holds true for the bipolar format.

### 3.1.2 FSM-Based Functions in SC

In SC, non-linear functions such as the hyperbolic tangent, sigmoid and exponentiation functions can be performed on stochastic bit streams using FSMs [29]. An FSM in SC can be viewed as a saturating counter that does not increment beyond its maximum value or decrement below its minimum value. For example, the FSM-based transfer function "Stanh" that approximates the hyperbolic tangent function is constructed such that

$$\tanh\left(\frac{na}{2}\right) \approx 2 \times \mathbb{E}[\text{Stanh}(n, \mathbf{a})] - 1, \tag{9}$$

where $n$ denotes the number of states in the FSM. Figure 2 illustrates the state transition of the FSM-based transfer function approximating the hyperbolic tangent function when using a set of states $\mathbb{C}_0 \to \mathbb{C}_{n-1}$. Since the sigmoid function is obtained from the hyperbolic tangent functions, the transfer function Stanh is also used to approximate the sigmoid function, that is,

$$\sigma(na) = \frac{1 + \tanh\left(\frac{na}{2}\right)}{2} \approx \mathbb{E}[\text{Stanh}(n, \mathbf{a})]. \tag{10}$$

### 3.2 Integral Stochastic Computing

In integral stochastic computing (ISC), a real value $s \in [0, m]$ in the unipolar format (or $s \in [-m, m]$ in the bipolar format) is represented as a sequence of integer numbers [30]. In this way, each element of the sequence $\mathbf{s} \in \{0, 1, \ldots, m\}^l$ in the unipolar format (or $\mathbf{s} \in \{-m, -m+1, \ldots, m\}^l$ in the bipolar format) is represented using the two's complement format, where $l$ denotes the length of the stochastic stream. The integral stochastic stream $\mathbf{s}$ is obtained by the element-wise additions of $m$ binary stochastic streams as follows

$$\mathbf{s} = \sum_{j=1}^{m} \mathbf{a}_j, \tag{11}$$

where the expected value of each binary stochastic stream, denoted as $\mathbf{a}_j$, is equal to $s/m$. With this definition, we have

$$\mathbb{E}[\mathbf{y}] = \sum_{j=1}^{m} \mathbb{E}[\mathbf{a}_j] = \sum_{j=1}^{m} \frac{s}{m} = s. \tag{12}$$

For instance, the element-wise addition of two binary stochastic streams, $\{0, 1, 1, 1, 1, 0, 1, 1\}$ and $\{0, 1, 1, 1, 0, 1, 1, 1\}$, each representing the real value of $0.75$, results in the integral stochastic stream

of $\{0, 2, 2, 2, 1, 1, 2, 2\}$ representing the real value of 1.5 for $m = 2$ and $l = 8$. We hereafter refer to the integral stochastic number generator function as ISNG.

Additions in ISC are performed using the conventional binary-radix adders, retaining all the input information as opposed to the scaled adders that decrease the precision of the output streams [30]. Multiplications are also implemented using the binary-radix multiplier in ISC. The main advantage of ISC lies in its FSM-based functions that take integral stochastic streams and output binary stochastic streams, allowing the rest of computations to be performed with simple bit-wise operations in binary SC. The approximate transfer function of hyperbolic tangent and sigmoid, which is referred to as IStanh, is defined as

$$\tanh\left(\frac{ns}{2}\right) \approx 2 \times \mathbb{E}[\text{IStanh}(n \times m, \mathbf{s})] - 1, \tag{13}$$

$$\sigma(ns) \approx \mathbb{E}[\text{IStanh}(n \times m, \mathbf{s})]. \tag{14}$$

The IStanh outputs zero when the state counter is less than $n \times m/2$, otherwise it outputs one. Considering $k_j$ as an entry of the state counter vector $\mathbf{k} \in \{0, 1, \ldots, m\}^l$ and $y_j$ as an entry of the IStanh's output vector $\mathbf{y} \in \{0, 1\}^l$, we have

$$y_j = \begin{cases} 0, & k_j < (n \times m/2) \\ 1, & \text{otherwise} \end{cases}, \tag{15}$$

where $j \in \{1, \ldots, l\}$. As opposed to the FSM-based functions in binary SC in which the state counter is incremented or decremented only by 1, the state counter of the FSM-based functions in ISC is increased or decreased according to the integer input value. In fact, the maximum possible transition at each time slot is equal to $m$ in ISC. Moreover, the FSM-based functions in ISC require $m$ times more states than the ones in SC. Despite the complexity of the FSM-based functions in ISC, they are more accurate than their counterparts in SC [30].

## 4 Synthesis of XNOR RNNs

### 4.1 Binarization of the Hidden State

In [20], the recurrent weights of LSTMs and GRUs were binarized using batch normalization in both the input-to-hidden and hidden-to-hidden transformations. More specifically, the recurrent computations of gate $\mathbf{f}_t$ is performed as

$$\mathbf{f}_t = \sigma\left(\text{BN}(\mathbf{W}_{fh}^b \mathbf{h}_{t-1}; \phi_{fh}, 0) + \text{BN}(\mathbf{W}_{fx}^b \mathbf{x}_t; \phi_{fx}, 0) + \mathbf{b}_f\right), \tag{16}$$

where $\mathbf{W}_{fh}^b$ and $\mathbf{W}_{fx}^b$ are the binarized weights obtained by sampling from the Bernoulli distribution as follows

$$\mathbf{W}^b = 2 \times \text{Bernoulli}(P(\mathbf{W} = 1)) - 1. \tag{17}$$

BN also denotes the batch normalization transfer function such that

$$\text{BN}(\mathbf{u}; \phi, \gamma) = \gamma + \phi \otimes \frac{\mathbf{u} - \mathbb{E}(\mathbf{u})}{\sqrt{\mathbb{V}(\mathbf{u}) + \epsilon}}, \tag{18}$$

where $\mathbf{u}$ is the unnormalized vector and $\mathbb{V}(\mathbf{u})$ denotes its variance. The model parameters $\gamma$ and $\phi$ determine the mean and variance of the normalized vector. The rest of the gate computations (i.e., $\mathbf{i}_t$, $\mathbf{o}_t$ and $\mathbf{g}_t$) are binarized in a similar fashion. So far, we have reviewed the method introduced in [20] to binarize the recurrent weights. We now extend this method to also binarize the hidden state vector $\mathbf{h}$. To this end, we use the sign function. However, the derivative of the sign function is zero during backpropagation, making the gradients of the loss w.r.t the parameters before the quantization function to be zero [17]. To address this issue, we estimate the derivative of the sign function as

$$\frac{\partial \text{sign}(\mathbf{h})}{\partial \mathbf{h}} \approx \begin{cases} 1, & |\mathbf{h}| < 1 \\ 0, & \text{otherwise} \end{cases}, \tag{19}$$

similar to [17]. In this way, the gradient's information are preserved. Training LSTMs with this method allows us to perform the matrix-vector multiplications of the gate computations using XNOR operations. We use the extended LSTM (ELSTM) with binary weights and the state hidden vector as our baseline for the rest of this paper.

## 4.2 Stochastic Representation of Gate Computations

Let us only consider the recurrent computations for a single neuron of a baseline's gate as

$$y = \alpha_h \sum_{j=1}^{d_h} w_h^j \odot h^j + \alpha_x \sum_{j=1}^{d_x} w_x^i \times x^j + b, \tag{20}$$

where $w_h$, $w_x$, $h$ and $x$ are the element entries of the hidden-to-hidden weight vector $\mathbf{w}_h \in \{-1, 1\}^{d_h}$, the input-to-hidden weight vector $\mathbf{w}_x \in \{-1, 1\}^{d_x}$, the hidden vector $\mathbf{h} \in \{-1, 1\}^{d_h}$ and the input vector $\mathbf{x} \in \mathbb{R}^{d_x}$, respectively. The bias is denoted as $b \in \mathbb{R}$. The parameters $\alpha_h \in \mathbb{R}$ and $\alpha_x \in \mathbb{R}$ denote the scaling factors dictated by the binarization process. Note that the batch normalization processes are considered in the parameters $\alpha_h$, $\alpha_x$ and $b$ in Eq. (20). In most of the temporal tasks, the input vector $\mathbf{x}$ is one-hot encoded, replacing the vector-vector multiplication of $\mathbf{w}_x \mathbf{x}$ with a simple indexing operation implemented by a lookup table. As such, let us merge this vector-vector multiplication into the bias as follows

$$y = \alpha_h \sum_{j=1}^{d_h} w_h^j \odot h^j + b. \tag{21}$$

Considering the linear property of the expected value operator, we can rewrite Eq. (21) as follows

$$y = \sum_{j=1}^{d_h} \alpha_h d_h \frac{w_h^j \odot h^j}{d_h} + b = \alpha_h d_h \mathbb{E}[\mathbf{w}_h \odot \mathbf{h}] + b = \mathbb{E}[\alpha_h d_h (\mathbf{w}_h \odot \mathbf{h}) + b] = \mathbb{E}[\mathbf{y}]. \tag{22}$$

So far, we have represented the output $y \in \mathbb{R}$ as a sequence of real numbers (i.e., $\mathbf{y} \in \mathbb{R}^{d_h}$) where each entry of the vector $\mathbf{y}$ is either $\alpha_h d_h + b$ or $-\alpha_h d_h + b$. Passing the vector $\mathbf{y}$ into the ISNG function generates the integral stochastic stream $\mathbf{y}^{ISC}$ such that

$$y = \mathbb{E}[\mathbf{y}] = \mathbb{E}[\text{ISNG}(\mathbf{y})] = \mathbb{E}[\mathbf{y}^{ISC}]. \tag{23}$$

Note that the integer range of the integral stream is equal to $\lceil |\alpha_h d_h| + |b| \rceil$. For instance, considering $\alpha_h = 0.2$, $d_h = 10$, $b = 0.5$ and $\mathbf{w}_h \odot \mathbf{h} = \{1, -1, 1, 1, 1, -1, 1, -1, -1, 1\}$, $\mathbf{y}^{ISC} = \{3, -2, 2, 3, 2, -1, 3, -1, -2, 2\}$ is an integral stochastic representation of $\mathbf{y} = \{2.5, -1.5, 2.5, 2.5, 2.5, -1.5, 2.5, -1.5, -1.5, 2.5\}$, resulting in $y = 0.9$. To guarantee the stochasticity of the sequence $\mathbf{y}^{ISC}$, we can permute the reading addresses of the memories storing the weights and the hidden state vector $\mathbf{h}$. Note that Eq. (20) with the input vector $x$ that is not one-hot encoded can also be represented as a stochastic stream by equalizing vector lengths of $d_x$ and $d_h$. Assuming that $d_h > d_x$ and $d_x$ is a multiple of $d_h$, this can simply obtained by repeating the input vector (i.e., $\mathbf{x}$) $d_h/d_x$ times as the mean of the repeated vector remains unchanged. Of course $d_h$ is a design parameter and can take any arbitrary value.

## 4.3 Weighted FSM-Based Function

So far, we have shown that the output of each neuron can be represented as an integral stochastic stream, allowing us to perform the nonlinear functions using the FSM-based IStanh function. However, our experiments show that the IStanh function fails to resemble the hyperbolic tangent and sigmoid functions (see Figures 3(a) and 3(b)). We attribute this problem to the even distribution of positive and negative integer elements in the vector $\mathbf{y}_{ISC}$ for both positive and negative values of $y$. More precisely, the vector $\mathbf{y}_{ISC}$ contains almost the same number of positive and negative integer entries since the expected value (i.e, the mean) of the vector $\mathbf{w}_h \odot \mathbf{h}$ is a small number. However, integral stochastic streams representing positive and negative real values are more likely to have more positive and negative entries, respectively. To address this issue, we propose a weighted FSM-based function, referred to as WIStanh, in which each state is associated with a weight. In the weighted FSM-based function, we use the same FSM that is used in the IStanh function. However, the output is determined by sampling from the weights associated to the states as follows

$$y_j^{\text{FSM}} = \text{Bernoulli}(\frac{w_{k_j} + 1}{2}), \tag{24}$$

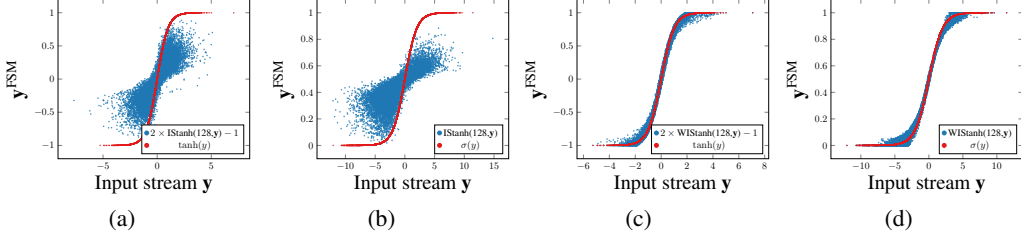

Figure 3: The IStanh function approximating (a) tanh and (b) sigmoid functions. The WIStanh function approximating (c) tanh and (d) sigmoid functions. The results were obtained by measuring the output of a single neuron for 12K input samples taken from the test set of the Penn Treebank dataset when performing character-level language modeling.

where $w_{k_j}$, $k_j$ and $y_j^{\text{FSM}}$ are entries of the weight vector $\mathbf{w} \in \mathbb{R}^{nm}$, the state counter vector $\mathbf{k} \in \{0, 1, \dots, m \times n - 1\}^{d_h}$ and the WIStanh's output vector $\mathbf{y}^{\text{FSM}} \in \{0, 1\}^{d_h}$ for $j \in \{1, \dots, d_h\}$. To obtain the weights approximating the FSM as the $\tanh$ function, we use linear regression such that

$$\tanh(y) = \sum_{q=0}^{m \times n - 1} p_{\mathbb{C}_q} \times w_q, \tag{25}$$

where $p_{\mathbb{C}_q}$ denotes the probability of the occurrence of the state $\mathbb{C}_q$ (i.e., the $q^{\text{th}}$ state in the state set of $\mathbb{C}_0 \to \mathbb{C}_{n \times m - 1}$). The sigmoid function can also be obtained in a similar fashion. Note that we constraint the weight values to lie into the interval of $[-1, 1]$. Figures 3(c) and 3(d) show the $\tanh$ and sigmoid functions implemented using the proposed WIStanh function where the FSM was trained on the Penn Treebank dataset [31] when performing the character-language modeling task. The early states of the trained FSM mainly contains values near to $-1$ in the bipolar format (or zero in the unipolar format) whereas the weight values of the latter states are close 1 (see Figure 4), complying with the state values of the conventional integral stochastic FSMs. Note that we fine tune the weights of our baseline model (i.e., ELSTM) with the proposed stochastic functions to comply with the approximation error.

## 4.4  XNOR LSTM

Let us rewrite the gate computations of LSTMs using the proposed stochastic representation as

$$\mathbf{F}_t^s = \text{WIStanh}(\text{ISNG}(\mathbf{W}_{fh}^b \mathbf{h}_{t-1}^b + \mathbf{W}_{fx}^b \mathbf{x}_t + \mathbf{b}_f)),$$
$$\mathbf{I}_t^s = \text{WIStanh}(\text{ISNG}(\mathbf{W}_{ih}^b \mathbf{h}_{t-1}^b + \mathbf{W}_{ix}^b \mathbf{x}_t + \mathbf{b}_i)),$$
$$\mathbf{O}_t^s = \text{WIStanh}(\text{ISNG}(\mathbf{W}_{oh}^b \mathbf{h}_{t-1}^b + \mathbf{W}_{ox}^b \mathbf{x}_t + \mathbf{b}_o)),$$
$$\mathbf{G}_t^s = \text{WIStanh}(\text{ISNG}(\mathbf{W}_{gh}^b \mathbf{h}_{t-1}^b + \mathbf{W}_{gx}^b \mathbf{x}_t + \mathbf{b}_g)), \tag{26}$$

where $\mathbf{F}_t^s \in \{0, 1\}^{d_h \times d_h}$, $\mathbf{I}_t^s \in \{0, 1\}^{d_h \times d_h}$, $\mathbf{O}_t^s \in \{0, 1\}^{d_h \times d_h}$ and $\mathbf{G}_t^s \in \{0, 1\}^{d_h \times d_h}$ denote the stochastic representation of the gate vectors (i.e., $\mathbf{f}_t$, $\mathbf{i}_t$, $\mathbf{o}_t$ and $\mathbf{g}_t$) in which each entry of the vectors is represented as a binary stochastic stream generated by the WIStanh function. More precisely, the expected value of the gate matrices $\mathbf{F}_t^s$, $\mathbf{I}_t^s$, $\mathbf{O}_t^s$ and $\mathbf{G}_t^s$ over their second dimension is equal to the gate vectors $\mathbf{f}_t$, $\mathbf{i}_t$, $\mathbf{o}_t$ and $\mathbf{g}_t$, respectively. This stochastic representation of the gates allows us to perform the Hadamard products of the state computations using XNOR operations. More precisely, we can formulate the state computations as

$$\mathbf{C}_t^s = \mathbf{F}_t^s \odot \text{SNG}(\mathbf{c}_{t-1}) + \mathbf{I}_t^s \odot \mathbf{G}_t^s, \quad \mathbf{h}_t = \text{S2B}(\mathbf{O}_t^s \odot \text{WIStanh}(\mathbf{C}_t^s)), \quad \mathbf{c}_t = \text{IS2B}(\mathbf{C}_t^s), \tag{27}$$

where $\mathbf{C}_t^s \in \{0, 1, \dots, m'\}^{d_h \times d_h}$ is an integral stochastic representation of the cell state vector $\mathbf{c}_t$. The SNG function generates a binary stochastic stream in the bipolar format (see Section 3.1), allowing us to replace the Hadamard products with XNOR operations. The S2B and IS2B functions convert binary and integral stochastic streams into a real number. In other words, these two functions find the expected value (i.e., the mean) of the stochastic streams by accumulating the stream entries and dividing the accumulated value by the stream length $l = d_h$. Of course setting the stream length

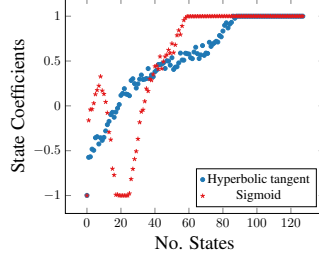

Figure 4: The weight values assigned for each state of the WIStanh to implement $\tanh$ and $\sigma$ functions.

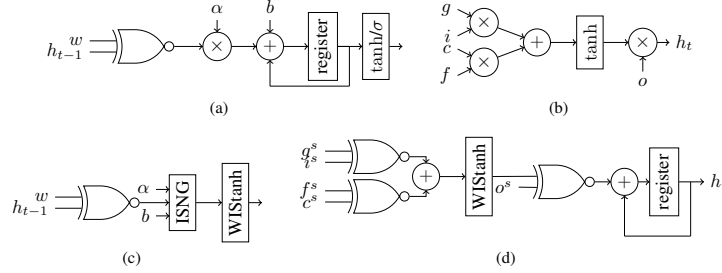

Figure 5: The main computational core of (a) the gate computations and (b) the state computations of the conventional binarized LSTM. The main computational core of (c) the gate computations and (d) the state computations in the proposed XNOR LSTM.

$d_h$ to a number of power of two replaces the division with a simple shift operation. With our stochastic representation, the accumulations of the vector-matrix multiplication at the gate computations are now shifted to the end of the state computations (see Figure 5). Moreover, both the gate and state computations now involve several vector-vector products, performed using XNOR operators. As opposed to stochastic computing systems requiring long latency to generate the stochastic streams, the stochastic bit streams of the proposed XNOR LSTM are already generated by the binarization of the gate computations. Therefore, the computational latency of the proposed XNOR LSTM is either the same or even less than of the conventional quantized LSTMs since using simple operators allows us to run the XNOR LSTM at higher frequencies.

## 5 Experimental Results

In this section, we evaluate the performance of the proposed XNOR LSTM across different temporal tasks including character-level/word-level language modeling and quation answering (QA). Note that the length of all stochastic streams (i.e., the parameter $l$) in our proposed method is equal to the size of LSTMs (i.e., the parameter $d_h$). For the character-level and word-level language modeling, we conduct our experiments on Penn Treebank (PTB) [31] corpus. For the character-level language modeling (CLLM) experiment, we use an LSTM layer of size 1,000 on a sequence length of 100 when performing PTB. We set the training parameters similar to [31]. The performance of CLLM models are evaluated as bits per character (BPC). For the word-level language modeling (WLLM) task, we train one layer of LSTM with 300 units on a sequence length of 35 while applying the dropout rate of 0.5. The performance of WLLM models are measured in terms of perplexity per word (PPW). For the QA task, we perform our experiment on the CNN corpus [32]. We also adopt the LSTM-based Attentive Reader architecture and its the training parameters, introduced in [32]. We measure the performance of the QA task as a error rate (ER). Note that lower BPC, PPW and ER values show a better performance. For a fair comparison with prior works, our XNOR-LSTM model for each task contains the same number of parameters as of their previous counterparts. Table 1 summarizes the performance of our XNOR-LSTM models. We consider a typical semi-parallel architecture of LSTMs, in which each neuron is implemented using a multiply-and-accumulate (MAC) unit, to obtain the implementation cost reported in Table 1. Depending on the precision used for the gate and state computations, we replace the multiplier inside the MAC unit with a simpler logic and report the implementation cost in terms of XNOR counts. In fact, we approximate the cost of a ternary/2-bit

Table 1: Performance of the proposed XNOR-LSTM models vs their quantized counterparts.

| | Baseline | LAB (ICLR'17 [19]) | AMQ (ICLR'18 [18]) | HitNet (NeurIPS'18 [26]) | ELSTM (ours) | XNOR (ours) |
|---|---|---|---|---|---|---|
| Precision of Gate Computations | FP | Binary | 2 bits | Ternary | Binary | Binary |
| Precision of State Computations | FP | FP | FP | FP | FP | Binary |
| **CLLM** Accuracy (BPC) | 1.39 | 1.56 | NA | NA | 1.47 | 1.52 |
| Size (MByte) | 16.8 | 0.525 | NA | NA | 0.525 | 0.525 |
| Cost (XNOR count) | 1,400,000 | 604,000 | NA | NA | 604,000 | 7,000 |
| Cost (No. clock cycles) | 1,000 | 1,000 | NA | NA | 1,000 | 1,000 |
| **WLLM** Accuracy (PPW) | 91.5 | NA | 95.8 | 110.3 | 93.5 | 95.5 |
| Size (KByte) | 2,880 | NA | 180 | 180 | 90 | 90 |
| Cost (XNOR count) | 420,000 | NA | 182,400 | 182,400 | 181,200 | 2,100 |
| Cost (No. clock cycles) | 300 | NA | 300 | 300 | 300 | 300 |
| **QA** Accuracy (ER) | 40.19 | NA | NA | NA | 40.4 | 43.8 |
| Size (MByte) | 7,471 | NA | NA | NA | 233 | 233 |
| Cost (XNOR count) | 1,433,600 | NA | NA | NA | 618,496 | 7,168 |
| Cost (No. clock cycles) | 256 | NA | NA | NA | 256 | 256 |

multiplication as two XNOR gates and the cost of a full-precision multiplication as 200 XNOR gates [17]. The experimental results show that our XNOR-LSTM models outperform the previous quantized LSTMs in terms of accuracy performance while requiring 86× fewer XNOR gates to perform the recurrent computations. While all the LSTM models in Table 1 require the same number of clock cycles to perform the recurrent computations, the inference time of our XNOR LSTMs is less than of other quantized works when running at higher frequencies due to their simpler operators. As a final note, the small gap between the XNOR and ELSTM models shows the approximation error caused by the use of stochastic computing.

## 6  Discussion

In Section 5, we only considered the implementation cost of our method in terms of XNOR operations since our main focus was to replace the costly multipliers with simple XNOR gates while the rest of the computing elements (i.e., the adders and look-up tables) almost remains the same (see Figure 5). Note that since SNG and ISNG can be easily implemented with magnetic tunnel junction (MTJ) devices which come almost at no cost compared to CMOS technologies [33], we excluded them from the implementation cost in Table 1. However, even if we include these units in our cost model, our stochastic-based implementation is still superior to its conventional binary-radix counterpart. To this end, we have implemented both the non-stochastic binarized method (e.g., [26]) and our proposed method on a Xilinx Virtex-7 FPGA device where each architecture contains 300 neurons. The implementation of our proposed method requires 66K FPGA slices while yielding the throughput of 3.2 TOPS @ 934 MHz whereas the implementation of the non-stochastic binarized method requires 1.1M FPGA slices while yielding the throughput of 1.8 TOPS @ 515 MHz. Therefore, our proposed method outperforms its binarized counterpart by factors of 16.7× and 1.8× in terms of area and throughput, respectively, while considering all the required logic such as SNG, ISNG and look-up tables. Note that the number of occupied slices denotes the area size of the implemented design. Also, the implementation of our proposed method runs at a higher frequency since its critical path is shorter than the conventional method due to the simpler hardware of XNOR gates versus multipliers. Therefore, this work is the first successful application of SC to the best of our knowledge where the SC-based implementation outperforms its conventional binary-radix counterpart in terms of both the computational latency and the area.

## 7  Conclusion

In this paper, we presented a method to synthesize XNOR LSTMs. To this end, we first represented the gate computations of LSTMs with binary weights and binary hidden state vector **h** in stochastic computing domain, allowing to replace the non-linear activation functions with stochastic FSM-based functions. We then proposed a new FSM-based function and its synthesis method to approximate the hyperbolic tangent and sigmoid functions. In this way, the gate activation values are represented as stochastic binary streams, allowing to perform the multiplications of the state computations using simple XNOR gates. To the best of our knowledge, this paper is the first to perform the multiplications of both the gate and state computations with XNOR operations.

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
