[Reviews · NeurIPS 2019]

Reviewer 1



Originality: This paper takes the next step for stochastic computing with LSTMs, quantizing all operations. It thus follows the obvious path to go and applies standard stochastic computing methods to get there. The novelity comes through the experimental analysis. Quality: There are no technical flaws. However, the evaluation metrics were defined in a way that they are skewed to show extraordinary benefits by neglecting some important contributions to the overall cost. Also, relevant newly introduced hyperparameters are arbitrarily chosen instead of evaluated. Clarity: The evaluations assume a hardware architecture for the baseline as well as their own work which is not described, thereby making some parts of the comparisons impossible to undestand in-depth. The required space for this could be obtained by shortening the 2-page primer on stochastic computing to a minimum. Some parameters (e.g. l) were chosen arbitrarily. Significance: Hard to say, because the presentation of the results does not allow for a comparison to non-stochastic computing methods. In the points compared, the results seem very promising.

Reviewer 2



This paper proposes an implementation of LSTMs that is believed to be more compatible with hardware chips by replacing the continuous non-linear operations of RNNs with logical operators based on XNOR gates and stochastic computation. The proposed WIStanh seems to be doing much better jobs in terms of approximating hyperbolic tangent function compared to IStanh function (baseline). I am a bit concerned by the binarization procedure and the trick that the authors used to flow the gradients : Eq. (19). Did the authors experience saturation issue by using this trick?

Reviewer 3



Con: Sentence structure and terminology is a bit unclear in some cases [e.g. line 69, 79] Some measures of comparison are not explained fully [e.g. Xilinx FPGA slices] Figure 3 is not very well explained [e.g. what was the sampling process, how many samples?] Could indicate best result in each row of Table 1 Pros: Novel approach Clear outline of the data set used for benchmarks Well-structured sections Good background section, well explained theory and mythology The experiments and comparisons made to other systems were well reasoned Motivation of the project was clearly outlined and justified Results reported show great decrease in computational complexity while achieving a better accuracy to similar models.

Reviewer 4



This paper proposes to use stochastic logic to approximate the activation function of LSTM. Binarization of non-linear units in deep neural networks is an interesting topic that can be relevant for low-resources computing. The main contribution of the paper was the application of stochastic logic to approximate activation functions(e.g. tanh). The authors applied the technique to a variant of LSTM model on PTB. Given that the technique is not really tied to the LSTM model, it would be more interesting to evaluate more model architectures(e.g. transformers), and compare them with the models that needs the non-stochastic logic versions. How would the approach compare to things like lookup table based the transformations? (given that we are already accumulating). Given that the PTB is a small dataset, which makes the result favorable for compression, would the approach generalize to bigger dataset?(e.g. wikipedia?). Direction of improvements: Having a dedicated evaluation of stochastic logic based activation will enhance the paper and allow the technique to be applied to a broader range of applications. From a practical point of view, implementing the method on a real hw(e.g. FPGA) will make the results more convincing, as the cost might goes beyond the number of gates. Finally, a major part of the paper is about designing the FSM for the stochastic logic activation. While this could be a contribution of the paper, it might be less relevant to the NeurIPS audiences.

[Author Response · NeurIPS 2019]

We thank the reviewers for their careful reading of our manuscript and their many insightful comments and suggestions towards
improving our paper. Below we provide a single response to all the comments of the reviewers, which will be added to the paper.
**Motivation**: The main motivation of this work is to propose **the first XNOR-LSTM model** where all the recurrent multiplications
in both the gate and the state computations are performed using XNOR operations. Note that the existing quantization methods (i.e.,
[18-19] and [26]) only focused on quantization of the gate computations while retaining the state computations in full-precision (FP).
**Originality**: To obtain an XNOR-LSTM model, we use stochastic computing in a substantially different way from the standard
stochastic computing (SC). Let us consider the vector-matrix multiplication of the gate computation as

8
$$\begin{bmatrix} h_{11} & h_{12} & \ldots & h_{1d_h} \end{bmatrix} \times \begin{bmatrix} w_{11} & w_{12} & \ldots & w_{1d_h} \\ w_{21} & w_{22} & \ldots & w_{2d_h} \\ \vdots & \vdots & \vdots & \vdots \\ w_{d_h1} & w_{d_h2} & \ldots & w_{d_hd_h} \end{bmatrix} \quad M = \begin{bmatrix} h_{11} \odot w_{11} & h_{11} \odot w_{12} & \ldots & h_{11} \odot w_{1d_h} \\ h_{12} \odot w_{21} & h_{12} \odot w_{22} & \ldots & h_{12} \odot w_{2d_h} \\ \vdots & \vdots & \vdots & \vdots \\ h_{d_h1} \odot w_{d_h1} & h_{d_h1} \odot w_{d_h2} & \ldots & h_{d_h1} \odot w_{d_hd_h} \end{bmatrix}.$$

In non-stochastic computing method, we simply perform the element-wise multiplication between the vector **h** and each column
of the matrix **W** to obtain the matrix **M**. Then, the accumulation over each column of the matrix **M** gives us the result of the
vector-matrix multiplication. This process is performed using a multiply-accumulate (MAC) unit on CPUs, GPUs and specialized
hardware. Having $d_h$ parallel MAC units, the vector-matrix multiplication takes $d_h$ clock cycles where $d_h$ denotes the number of
rows and columns of the square weight matrix **W**. In the standard SC, a binary stochastic stream of size $l$ is generated for each
element of both the vector **h** and the matrix **W**, introducing an additional dimension of size $l$ to them and an overhead latency of $l$
clock cycles. For example, the standard stochastic version of the vector **h** is a matrix of size $d_h \times l$. Therefore, even though the
standard SC allows to perform the vector-matrix multiplication using XNOR operations, it suffers from the long computation time
overhead (see [20] and [28]). In our work, however, we took substantially a different approach. The main idea was started with this
question: Can we treat the row of **h**, each column of **W** and consequently each column of **M** as stochastic streams of length $l = d_h$ if
all the elements were binary? In this way, **we do not generate any stochastic stream and we only treat each column of the M as**
**a stochastic stream**. Compared to the non-stochastic computation, we only perform the element-wise multiplication without any
accumulation over the columns of **M**, allowing us to perform the state computations using stochastic logic units (i.e., in binary).
Note that since there is always a scaling factor $\alpha$ in the binarization process and bias, we tweak our representation from binary SC to
integral SC. We then proposed an integral SC tanh function that takes each column of the matrix **M** and returns a binary stochastic
stream of the same length, approximating the non-linear functions used in LSTMs. Now, we have the gate values (i.e., $f$, $i$, $o$ and
$g$) represented as binary stochastic stream, allowing us to replace the multiplications in Eq. (2) with XNOR operations. When the
state computations are done, we perform accumulation over the stochastic streams to obtain real values of the next state vector **h**. In
fact, compared to the conventional binarized LSTM models (e.g., [26]) as shown in Figure (a) and (b), the accumulator unit in the
gate computations of the conventional method is shifted to the end of state computations in our stochastic computing method (see
Figure (c) and (d)). Note that the **length of all stochastic streams** (i.e., **the parameter** $l$) in our proposed method is equal to **the**
**size of LSTMs** which is a design parameter and **denoted as** $d_h$ in the paper. To binarize the weight matrix **W** and the hidden state
vector **h**, we leveraged the non-SC techniques introduced in [17] and [20] as described in Section 4.1. Note that sampling from the
Bernoulli distribution in Section 4.1 only happens during the training phase to obtain binarized weights. Once the training is finished,
deterministic binary values are stored for inference and we treat these deterministic binary values as stochastic streams in our work.
Therefore, both weights and hidden state values are stored as deterministic binary values, reducing the memory footprint by a factor
of $32\times$ compared to FP. Moreover, the number of I/O and memory elements are the same as of conventional quantization methods
since we only viewed the binarized weights and hidden states as stochastic streams.
**Implementation Cost**: In the comparison section, we only compared the cost of our method in terms of XNOR operations since our
main focus was to replace the costly multipliers with simple XNOR gates while the rest of computing elements (i.e., the adders
and look-up tables) almost remains the same (see Figure (a,b,c,d)). Note that since SNG and ISNG can be easily implemented
with magnetic tunnel junction (MTJ) devices which come almost at no cost compared to CMOS technologies, we excluded them
from the implementation cost. However, based on the reviewers' comment, we have implemented both non-stochastic binarized
method (e.g., [26]) and our proposed method on a Xilinx Virtex-7 FPGA device where each architecture contains 300 neurons. The
implementation of our proposed method requires 66K FPGA slices while yielding the throughput of 3.2 TOPS @ 934 MHz whereas
the implementation of the non-stochastic binarized method requires 1.1M FPGA slices while yielding the throughput of 1.8 TOPS @
515 MHz. Therefore, our proposed method outperforms its binarized counterpart by factors of **16.7×** and **1.8×** in terms of **area** and
**throughput**, respectively, while **considering all the required logic** such as SNG, ISNG and look-up tables. Note that the number
of occupied slices denotes the area size of the implemented design. Also, the proposed implementation runs at a higher frequency
since its critical path is shorter than the conventional method due to the simpler hardware of XNOR gates vs multipliers.
**WikiText-2**: Based on the reviewer's comment, we have performed our method on WikiText-2 dataset which contains 33K vocabulary
and is $3\times$ larger than PTB. We obtained PPW values of 105.5, 107.3 and 109.4 for FP baseline, our ELSTM model and our XNOR
model on a hidden size of 512 (i.e., $d_h = 512$), respectively. The obtained results are consistent with the results obtained for PTB.
**Figure 3**: To obtain the results in Figure 3, we measured the output of a single neuron for 12K input samples taken from the test set
of PTB when performing CLLM.
**Significance**: In this work, we presented a stochastic computing method that enables us to perform all the recurrent multiplications
using XNOR operations. We believe that the proposed technique can be introduced to NeurIPS audiences with a successful
application to quantization of LSTMs which is of a paramount importance when designing dedicated hardware. We also agree with
the reviewer's comment that the proposed stochastic method is a general approach and can be used in other applications, making it
even more interesting to NeurIPS audiences. Moreover, we believe that this work will have a huge impact on the SC community as
this is **the first successful application of SC** where using SC preserves the latency intact as apposed to the standard SC that incurs a
long latency when comparing with the non-stochastic implementations.

(a) Non-stochastic Gate Computations   (b) Non-stochastic State Computations   (c) Our Stochastic Gate Computations   (d) Our Stochastic State Computations

[Meta-Review · NeurIPS 2019]

This paper introduces interesting stochastic finite state machine based methods to approximate nonlinear activation functions including hyperbolic tangent and sigmoid functions. A fully binary model of LSTM (both weights and hidden states are binary) is constructed in which XNOR operations are used to perform all the multiplications in the gate and state computations. Empirical results show that the proposed binary LSTM model can dramatically reduce the computational lost while without sacrificing latency or accuracy comparing with existing methods. In the rebuttal, concerns from the reviewers are carefully addressed, e.g., adding an FPGA based implementation. However, some of them are still lack of sufficient details and discussions, in particular, the cost of stochastic computing, and the memory movement cost. In addition, this paper only considered 1-layer LSTM. It is unclear how the proposed method can be generalized to other architectures. We will be happy to see these concerns are addressed in the revised version of this paper.